# Non-Coding RNAs in COVID-19: Emerging Insights and Current Questions

**DOI:** 10.3390/ncrna7030054

**Published:** 2021-08-31

**Authors:** Tobias Plowman, Dimitris Lagos

**Affiliations:** 1York Biomedical Research Institute, University of York, Wentworth Way, York YO10 5DD, UK; tobias.plowman@york.ac.uk; 2Hull York Medical School, University of York, Wentworth Way, York YO10 5DD, UK

**Keywords:** COVID-19, non-coding RNA, microRNA, lncRNA, inflammation, cytokine storm syndrome, vascular regulation, haemostasis, RNA–RNA interactions

## Abstract

The highly infectious severe acute respiratory syndrome coronavirus 2 (SARS-CoV-2) emerged as the causative agent of coronavirus disease 2019 (COVID-19) in late 2019, igniting an unprecedented pandemic. A mechanistic picture characterising the acute immunopathological disease in severe COVID-19 is developing. Non-coding RNAs (ncRNAs) constitute the transcribed but un-translated portion of the genome and, until recent decades, have been undiscovered or overlooked. A growing body of research continues to demonstrate their interconnected involvement in the immune response to SARS-CoV-2 and COVID-19 development by regulating several of its pathological hallmarks: cytokine storm syndrome, haemostatic alterations, immune cell recruitment, and vascular dysregulation. There is also keen interest in exploring the possibility of host–virus RNA–RNA and RNA–RBP interactions. Here, we discuss and evaluate evidence demonstrating the involvement of short and long ncRNAs in COVID-19 and use this information to propose hypotheses for future mechanistic and clinical studies.

## 1. SARS-CoV-2 and COVID-19

Severe acute respiratory syndrome coronavirus 2 (SARS-CoV-2) is the most recently discovered human-infectious and pathogenic coronavirus (CoV) that is thought to have emerged in December 2019 in China [1,2]. It is the causative agent of coronavirus disease 2019 (COVID-19). SARS-CoV-2 is an enveloped positive-sense single-stranded RNA virus from the *betacoronavirus* subfamily and is approximately 80% identical in its nucleotide sequence to severe acute respiratory syndrome coronavirus (SARS-CoV) that caused a more short-lived pandemic in 2002–2003 [2]. Similar to SARS-CoV, SARS-CoV-2 binds the human angiotensin converting enzyme 2 (ACE2) receptor to gain host cell entry, focusing infection on respiratory cells [3], although other human ACE2-expressing cell types can be affected [4,5].

SARS-CoV-2 infection can lead to a wide range of outcomes from asymptomatic infection to life-threatening lung disease alongside peripheral disorders [3]. Severely affected patients present with acute respiratory distress syndrome (ARDS) from lung damage, thromboembolic disorders, cardiovascular, cardiac and gastro-intestinal dysregulation, and/or liver or kidney malfunction [6]. As a result, COVID-19 has a high mortality rate that is estimated to be around 1% of cases [7,8,9] and has the capacity to overwhelm healthcare systems if left to spread uncontained in a population. Mortality risk also increases sharply with associated risk factors, such as age, autoimmune conditions such as diabetes [10], and existing cardiovascular or chronic lung disease [11]. 

## 2. The Human Non-Coding Genome and COVID-19

The nuances in an organism’s genome were traditionally appreciated for the specific forms of proteins that it produced. According to this paradigm, a gene is transcribed to produce the corresponding RNA, and the translational machinery converts this nucleotide blueprint into an amino acid sequence with biological function, with differing sequences conferring altered functions that are helpful to the specific organism. However, transcribed RNA is not always translated, and it itself can have biological function without encoding protein, known as non-coding RNA (ncRNA).

Forms of ncRNA can be sub-classified based on length. Transcripts longer than 200 base pairs (bp) are deemed ‘long non-coding RNAs’ (lncRNAs), and those shorter than 200 bp are designated as small ncRNAs, such as microRNAs (miRNAs) or small nucleolar RNAs (snoRNAs). There further subdivisions exist, for instance lncRNAs encoded from intergenic regions are deemed lincRNA, and miRNAs with the 5′ and 3′ termini end joined together are called circular RNAs (circRNAs).

The non-coding portion in simple eukaryotes is between 25–50% of the genome; in plants and complex fungi, this number is generally between 50 and 75%, whilst in humans, an incredible 98.5% of the genome is non-coding [12]. Given that the majority of the mammalian genome is indeed transcribed [13], not only does this imply that ncRNAs may be key in facilitating the development of complex organisms, it may also relate to the ability of humans to produce multifaceted immune responses that are formulated from a cascade of integrated signalling events. Conversely, the genomically simpler parasitic by-products of the genetic code, viruses, also have the capacity to produce ncRNAs, utilising endogenous mechanisms to promote their propagation.

Until recent decades, the relevance of ncRNAs has been overlooked in biology. However, predictably, given their genomic prevalence, findings continue to demonstrate their involvement in key cellular processes. For instance, the lncRNA *Xist* is the initiator of X-chromosome inactivation in female cells, a fundamental developmental process within which many mechanisms were a mystery until the incorporation of ncRNA biology [14,15]. In this context, the 17 kb-long *Xist* is thought to behave as a recruiting platform for the repression factors that mediate the silencing of gene expression from the X-chromosome [15]. One lincRNA, MALAT1, acts as a linker or scaffold structure to position nuclear speckles at gene loci that are being transcribed to aid in splicing [16], though its action in other immunological contexts has also been described [17]. Particularly intriguing research into the miRNA miR-122 uncovered it as a nonredundant host factor that promotes hepatitis C virus (HCV) infection. It, together with Argonaute protein 2, protects the HCV genome from degradation, stabilising HCV RNA, thus prolonging cellular infection [18].

Emerging evidence shows that both host- and virus-derived ncRNAs also play key roles in protection from and susceptibility to severe COVID-19. Studies continue to elucidate differential ncRNA expression profiles in patients with COVID-19, in in vitro cell lines, and in animal models. These profiles are useful for providing a basis for stratifying disease states, therefore acting as biomarkers. In addition, growing interest in the effects of SARS-CoV-2-originating RNA in remodelling the behaviour of host RNA, proteins, and signalling pathways highlight several possible detrimental outcomes from infection. Therefore, despite the infancy of the ncRNA-COVID-19 field, the pace at which data and publications are released warrants a novel aggregation to connect the conclusions presented in a mechanistically meaningful way and to form hypotheses to encourage new investigations into ncRNAs in COVID-19. A brief summary of the major ncRNAs identified in patient profiling studies is shown in Table 1.

## 3. ncRNAs in Cytokine Storm Syndrome, a Hallmark of Severe COVID-19

### 3.1. COVID-19 and Cytokine Storm Syndrome

Patients with severe COVID-19 routinely present both observational and laboratory features of a cytokine storm syndrome (CSS), a term that groups several hyperinflammatory disorders [59,60]. Broadly, these conditions are characterised by extreme cytokine production as a result of immune hyperactivity [59]; however, each form is a unique amalgamation of inflammatory symptoms. For instance, COVID-19 presents with fever, ARDS, and elevated immune marker levels in sera, such as interleukin (IL-6) [61,62]. Patients also regularly exhibit lymphocyte-specific cytopenia [1] and the recruitment of immune cells to the lung [63]. Likewise, another form of CSS, haemophagocytic lymphohistiocytosis (HLH), presents with fever and cytopenias and the accumulation of lymphocytes and macrophages in tissues such as the spleen, bone marrow, and liver [64]. Therefore, although COVID-19 resembles other types of CSS, its differences mean efforts must be undertaken to understand its novel immunopathological landscape for the identification of relevant treatment.

### 3.2. COVID-19 Loss of miR-1275 and miR-766-3p may Promote an IL-6 Hyperinflammatory State

A cytokine regularly highlighted for its importance in CSS is IL-6, a pleiotropic molecule whose signalling is targeted therapeutically in several inflammatory disease contexts, for instance, in rheumatoid arthritis [20] and, notably, in the CSS condition HLH [65]. Despite the unknowns surrounding immunopathology in COVID-19, strides have already been made to implicate ncRNAs in the regulation of several related immunomodulatory signalling pathways. For instance, a profiling study uncovered several intriguing differential miRNA expression signatures between ten COVID-19 patients and ten healthy control individuals [19]. First, miR-1275 was significantly downregulated in the COVID-19 patient cohort (Log_2_FC = −5.48), a miRNA implicated in NF-κB IL-6 regulation. For instance, a separate study noted that the treatment of adipocytes with tumour necrosis factor alpha (TNF-α) and IL-6 reduced miR-1275 expression and that the miR-1275 gene promoter harbours NF-κB-binding sites, allowing transcriptional repression after NF-κB activation [66]. Therefore, miR-1275 may be a useful biomarker in the identification of an inflammatory state in COVID-19. Further, the most statistically significant differentially expressed miRNA in the COVID-19 profiling study was miR-766-3p, which was downregulated in individuals with COVID-19 [19]. This miRNA has been previously shown to also be significantly downregulated in the sera of rheumatoid arthritis patients, where patients also exhibit upregulated IL-6 [21]. The authors further investigated this relationship by artificially modulating miR-766-3p expression in MH7A cells, a human arthritis synovial cell line, after an immune stimulation event. They exposed peripheral blood mononuclear cells (PMBCs) to lipopolysaccharide (LPS), co-cultured these PBMCs with MH7A cells, and subsequently measured IL-6 production. They found that miR-766-3p overexpression significantly reduced IL-6 expression in the context of immunological stimulation [21]. The immunomodulatory NF-κB signalling pathway is heavily implicated in this ncRNA-related inflammation, as this same study showed a reduction of expression from an NF-κB promoter-driven luciferase reporter with LPS stimulation after miR-766-3p transfection [21]. As such, in COVID-19 patients, miR-766-3p reduction may represent a biomarker of increased IL-6 expression or, more interestingly, may itself have anti-inflammatory function. Given the experimental data that its overexpression reduces IL-6 induction in immunological stimulation, miR-766-3p may represent a node of regulatory integration, where its loss promotes an IL-6 over-expression inflammatory state after factoring in of other immunomodulatory signals. This miRNA may represent a key druggable or modifiable example of ncRNA-related crosstalk in the NF-κB and IL-6 axis.

A further comparison in the miRNA profiling study [19] stratified COVID-19 patients based on the need for oxygenation therapy, and the significant downregulation of miR-766-3p remained in both groups. Patient categorisation based on the need for oxygenation is a common method to classify COVID-19 severity, this aiding in uncovering of the key mechanisms that characterise the worsening of the disease [67,68]. However, whilst usually correlated with known disease events in COVID-19, it may be not perfectly predictive of specific biomolecular expression levels on an individual patient basis. Moreover, the reduction of miR-766-3p was more substantial in those who did not require oxygenation (log2FC = −4.19 in un-oxygenated, −2.78 in oxygenated) [19], conflicting with the observation that IL-6 levels are generally higher in those with severe COVID-19 [68,69] and so should expectedly correlate with lower miR-766-3p, thus resulting in worse disease.

These data and the interpretations made by the authors represent a key challenge in unravelling the complex and fast-moving picture that characterises ncRNA-based regulation and the immunopathological events of COVID-19 in addition to the difficulties in ensuring key data are available for interpretation. The data in study [19] are convincing to implicate miR-766-3p in IL-6 expression and thus CSS, particularly when placed within the context of the existing literature. However, further and repeated study with larger datasets, after stratifying for COVID-19 severity and/or serum cytokine expression levels, would be justified in considering miR-766-3p aw key molecule in this inflammatory event mediated directly by IL-6 in CSS. This may culminate in the establishment of a clinically useful biomarker or even present as a target for drug discovery.

### 3.3. miR-146a-5p Expression Is a Determinant of Response to Therapeutic IL-6R Blockade

As early as February of 2020, the anti-IL-6 receptor (IL-6R) blocking antibody tocilizumab was assessed for therapeutic efficacy in a randomised control trial in China [70] to counter the putative COVID-19-induced CSS, where IL-6 is thought to play a key role. With varying degrees of efficacy, several large-scale clinical trials have now demonstrated a reduction in the mortality rate or symptomatic improvement in COVID-19 patients by blocking IL-6 signalling [68,71]. One serum miRNA profiling study [25] incorporated miR-146a-5p not only into COVID-19-related CSS, but also in the responsiveness of patients to CSS-countering treatment where IL-6 is the targeted cytokine.

After dividing a cohort of COVID-19 patients into those who tocilizumab-responsive (*n =* 16) and those who were unresponsive (*n =* 13), the authors retrospectively established no baseline difference in miR-146a-5p expression between the groups [25]. However, the COVID-19 patients expressed lower miR-146a-5p levels compared to the healthy controls. Intriguingly, after treatment, those in the ‘tocilizumab responder’ group exhibited a significant increase in miR-146a-5p expression, whilst those unresponsive conversely experienced a significant decrease in the expression of the miRNA. Levels of miR-146a-5p also inversely correlated with post-treatment IL-6 serum levels, and those with the lowest expression levels of miR-146a-5p had the most adverse outcomes. As such, in this instance, whilst miR-146a-5p as a pre-treatment biomarker appears to not be predictive of responsiveness to therapy, its expression changes from treatment are linked with IL-6 expression and adverse outcomes in patients. However, is its only utility as a biomarker as an output of IL-6 expression and treatment success? Or does it participate in the regulation of this inflammatory axis?

Though unclear from this study [25], an inverse relationship where miR-146a-5p lies upstream of IL-6 expression has been frequently established in the literature (Figure 1a). Inhibition of miR-146a in LPS-stimulated cystic fibrosis macrophages increased IL-6 production [72], and m iR-146a overexpression in human retinal endothelial cells reduced IL-6 expression [26]. In further models, an IL-1β-stimulated murine cementoblast line exhibited increased in IL-6 expression that was significantly enhanced upon miR-146a-5p inhibition [73], and miR-146a knockout mice expressed high levels of IL-6 in sera [74]. It is thought that miR-146a exerts its immunosuppressive effect by targeting IRAK1 [27], thus inhibiting NF-κB-responsive gene expression [75]. These observations suggest that therapeutic efficacy may depend on an increase in miR-146a-5p, suggesting a host response in the form of upregulated miR-146a-5p is necessary to complement the therapy-mediated reduction in IL-6 signalling.

Endogenous increases to miR-146a expression are thought to result from the binding of the promoters NF-κB and c-Myc, whilst reduced expression is attributed to DNA methylation at CpG sites in promoter regions [29]. Intriguingly, the lincRNA PVT1 is implicated in inducing miR-146a promoter CpG methylation and subsequent repression, representing an example of ncRNA-based crosstalk in immunomodulation [29]. In a separate study, PVT1 was shown to be significantly upregulated (FC = 3.62, FDR = 0.004956) in the bronchoalveolar lavage (BAL) samples of 12 COVID-19 patients compared to healthy controls [28]. PVT1 has also previously been implicated in inflammatory disease, for instance, upregulated in the synovial tissue of rheumatoid arthritis patients [30]. Therefore, the upregulation of PVT1 may act upstream of miR-146a, inducing its downregulation, thus leading to enhanced cytokine production in an inflammatory setting. Further exploration of the role of this ncRNA network of crosstalk in COVID-19 inflammation appears highly promising.

Interestingly, the protective effects of miR-146a upregulation may have been demonstrated in a study that looked at the relatively unexplored effects of COVID-19 in pregnant women [76]. First, miRNA expression was found to be correlated with blood samples and those collected from the placenta at delivery, emphasising the relevance of using findings from sera to reflect upon the inflammatory state of diverse tissues. It was then determined that despite systemic and placental increases in inflammation markers, such as IL-1β and IL-6, little to no symptoms of COVID-19 presented. Alongside the noted upregulation of suspected ‘anti-viral’ miRNAs such as miR-21, miR-28, and miR-98, the immunoregulatory miR-146a may have played its part in subduing the immune response in these SARS-CoV-2-positive women.

### 3.4. Upregulation of Inflammatory miR-155 May Be Combatted by Glucocorticoid Treatment

An intriguing upregulation of the miRNA miR-155 was noted in sera specific to severe COVID-19 patients (*n =* 18) when compared to healthy controls (*n =* 15) and also in a separate COVID-19 patient cohort (*n =* 20) compared to patients with influenza-induced ARDS (*n =* 13) [52]. This miRNA may therefore represent part of the immunopathological picture uniquely found in cases of COVID-19.

This miRNA has been discussed as a ‘master of inflammation’, with intriguing links to COVID-19-associated pathology. In inflammatory responses, miR-155 regulates NF-κB signalling and may coordinate with miR-146a in immunomodulation [56]. It is thought that miR-155 acts antagonistically to miR-146a, as the artificially elevated expression of miR-155 nullified the immunosuppressive action of miR-146a in terms of NF-κB activation in the macrophages of transgenic mice [77]. A proposed model suggests that miR-155 expression is the first step in the immune activation cascade, which is upregulated by NF-κB signalling, and feeding back via the IKK signalosome complex and PI3K/Akt to further amplify NF-κB [77] (Figure 1b). In a second stage, the NF-κB-driven expression of miR-146a works to negate NF-κB activation by modulating IRAK1 [27,77]. Given miR-146a upregulation is implicated in successful response to anti-IL-6 CSS blockade and that miR-155 upregulation is unique to COVID-19 and not severe influenza, this miRNA regulatory cascade may be a core inflammatory axis in SARS-CoV-2-related disease whose dysregulation is central to its unique disease context. Indeed, in the aforementioned study in pregnant women with COVID-19 [76], miR-155 was also upregulated, further suggesting that a unique balance of miRNA expression may be the difference between the extremes of severe and asymptomatic COVID-19.

Levels of miR-155 may also be relevant to one of the most successful COVID-19 therapies to date, glucocorticoids, specifically dexamethasone. Dexamethasone is now first-line treatment in several countries owing to success from large scale clinical trials [67,78]. In a study of LPS-activated cultured macrophages, dexamethasone inhibited the expression of miR-155, and inhibition of miR-155 using an oligonucleotide inhibitor achieved a similar therapeutic effect in terms of reducing IL-6, TNF-α, and NO production [79]. Further, the study elucidated this mechanism to act via glucocorticoid receptors, which inhibited NF-κB activation [79]. Given that the expression of miR-155 is dependent on NF-κB binding to response elements in its promoter, dexamethasone treatment thus indirectly inhibited miR-155 production. There is no literature to date investigating this specific miRNA-based mechanism in COVID-19. However there are some provocative links that deem further investigation, including that miR-155 is 1) upregulated in a cohort of COVID-19 patients [52]; 2) strongly implicated in NF-κB signalling [56,77]; 3) a coordinator of inflammation in concert with miR-146a [77] (which itself is seemingly highly relevant in indicating therapeutic success in COVID-19 [25]); and 4) specifically involved in tilting the axis towards immunosuppression in glucocorticoid treatment [79], a first-line treatment with proven clinical success in COVID-19 [67,78]. It may be interesting, for example, to perform a similar study to that which compared miR-146a levels after tocilizumab treatment [25].

## 4. ncRNAs in Immune Cell Recruitment and Differentiation, a Precursor to Damage

### 4.1. The Infiltration of Immune Cells Is Associated with Worsened COVID-19

COVID-19 severity is correlated with a related paradigm of immunological change: the infiltration of innate and mesenchymal-derived immune cells to the lung microenvironment [63]. It is both contributory to CSS and the effector arm of CSS, responding to the secretion of chemokines, and is responsible for producing damaging molecules. For instance, more severe COVID-19 correlated with increases to the neutrophil-to-lymphocyte ratio [80]. Cell profiling studies on BAL samples in severely affected patients also identified mononuclear phagocytes as constituting 80% of the total cell population [81]. In mildly affected patients, this proportion was around 60%, and in healthy controls, it was around 40%. Additionally, the noted upregulation of CCL2, CSCL10, and GM-CSF in seriously affected patients may induce the taxis of monocytes and neutrophils to the lung compartment [63]. The general effect of this recruitment is function-compromising lung damage. For instance, neutrophilia in COVID-19 patients is associated with lung lesion progression, which is likely an effect of the enhanced secretion of neutrophil extracellular traps (NETs) [82]. NETs function to combat infection through the action of oxidative enzymes and complexes of DNA, histones, and antimicrobial proteins; however, NETs damage tissues and can kill endothelial cells [82].

### 4.2. Differential miR-31-5p Expression May Impact Immune Cell Recruitment

Another differentially expressed miRNA in the previously mentioned profiling study [19] was miR-31-5p, which was significantly and substantially upregulated in the COVID-19 patient cohort (Log_2_FC = 6.51). In psoriasis keratinocytes from patients (inflammatory skin disease), miR-31-5p was overexpressed, and its silencing suppressed keratinocyte activation and their ability to recruit immune cells via the expression of CXCL1, CXCL5, and CXCL8 [22]. Further, a predicted target gene of miR-31-5p, serine/threonine kinase 40 (STK40), represses NF-κB signalling, and the effects of miR-31-5p expression were lost when STK40 was silenced [22]. Therefore, in COVID-19 patients, miR-31-5p upregulation may inhibit the immunosuppressive activity of STK40, permitting chemokine ligand expression, enhancing inflammation and immune cell recruitment. However, miR-31-5p, has also been implicated in activity that would ordinarily be protective in COVID-19, as one study predicted one of its targets to be Tensin-1 (TNS1), a contributory factor to enhanced immune cell infiltration [23], associating miR-31-5p upregulation with TNS1 downregulation in patients with colon adenocarcinoma. Similarly, miR-31 knockdown, though upregulated in patients with ulcerative colitis and Crohn’s disease, was shown to worsen inflammation in a mouse inflammatory bowel disease model, as expression of IL-7R, IL-17RA, and IL6ST increased [24]. Therefore, this may be evidence of the unique ncRNA immunopathological picture that depicts COVID-19, as miRNAs that have had pro-inflammatory or anti-inflammatory roles in human disease may instead form a component of a complex integration of the immunomodulatory signals involved in immune cell recruitment.

### 4.3. MALAT1 Expression Is Altered in SARS-CoV-2 Infection with Inflammatory Implications

Other types of ncRNAs have also been implicated in pathogenic immune cell recruitment. One study showed expression changes in four snoRNAs alongside pseudogenes that likely have regulatory function after the infection of normal bronchial epithelial (NHBE) cells with SARS-CoV-2 [83]. One study established that COVID-19 induces expression changes to 114 circRNAs in the peripheral blood of patients, which enrichment analysis deemed to be related to the regulation of the cell cycle and inflammation [84]. In another report, SARS-CoV-2 infection upregulated 155 lncRNAs and downregulated 195 lncRNAs in NHBE cells [31]. One lincRNA of interest, MALAT1, was upregulated [31]. Previously, it was shown that MALAT1 silencing in a lung transplant ischemia-reperfusion rat model inhibited neutrophil chemotaxis by recruiting p300 and reducing IL-8 expression [32]. Therefore, these data indicate that MALAT1 may also be upregulated in COVID-19 patient lung cells, encouraging the taxi of immune cells and subsequent damaging inflammation.

Changes to MALAT1 expression are a recurring observation in COVID-19 lncRNA profiling studies and infection models. One study currently in preprint compared lncRNA expression in several cell types from BAL and PBMC samples from healthy, mildly affected, and severely affected COVID-19 patients [33]. In BAL samples, MALAT1 was under-expressed in monocytes and macrophages in mild patients, compared to healthy and severely affected patients. This may contribute to the ability of some patients to avoid pathological inflammation, as MALAT1 expression is implicated in the activation and maturation of macrophages into their classical M1 subtype [34].

Conversely, MALAT1 suppression is thought to be integral to CD4+ T cell activation and differentiation towards an inflammatory Th1 and Th17 effector type [35]. In a mouse study, Malat1 knockouts in Th1 and Th2 cells resulted in lower the expression of the immunosuppressant IL-10 and more robust immune responses in infection [17]. In the COVID-19 BAL samples, CD4+ T cells overexpressed MALAT1 in mild patients and under-expressed it in severe cases [33]. As MALAT 1 loss also pushes CD4+ T cells away from a regulatory T cell subtype [35], it may suggest that severe patients exhibit a hyperinflammatory response in part due to MALAT1 loss in CD4+ T cells, resulting in the secretion of pro-inflammatory cytokines. The reverse may be true for mildly affected patients who are exhibiting potentially protective MALAT1 overexpression in CD4+ T cells.

The study’s most significantly differentially expressed gene was *NEAT1* [33], which encodes a lincRNA in genomic proximity to MALAT1. NEAT1 has been shown to promote inflammation by enhancing the assembly and processing of inflammasomes [36]. NEAT1 was overexpressed in nine cell types identified from severe COVID-19 patient BAL samples, including in M1 and M2-type macrophages, monocytes, CD4+ T cells, and CD8+ memory T cells [33]. NEAT1 was not differentially expressed in PBMCs, indicating this immunological effect is specific to the lung, i.e., the site of infection and inflammation. Interestingly, the NEAT1 in the BAL cell types was generally under-expressed in mild samples compared to healthy samples, suggesting that a mechanism of NEAT downregulation may also protect mildly affected COVID-19 patients. The above observations support further studies to dissect the role of MALAT1 and NEAT1 in COVID-19 immunity and pathogenesis.

### 4.4. miRNA-Related Endothelial Dysfunction in the Lung Microenvironment

Endothelial cell behaviour is key as the initiator of immune cell recruitment and subsequent inflammation; therefore, their function in this context may be directly related to the likelihood of COVID-19 disease. In a targeted study, fixed post-mortem lung tissue from COVID-19 patients that suffered severe respiratory injury and/or thrombotic events were analysed for their expression of miR-26a-5p, miR-29b-3p, and miR-34a-5p [37]. These miRNAs are implicated in endothelial cell function, inflammation, and viral disease, so they encapsulate critical features of COVID-19 pathogenesis. In the study, immunohistochemistry-inferred expression of cytokine storm-related proteins allowed several intriguing correlations to be identified. All three miRNAs were significantly downregulated in the analysed COVID-19 patient lung samples (*n =* 9) compared to the control group (*n =* 10). Significant independent inverse correlations were then identified between miR-29b-3p and IL-4 and IL-8 as well as between miR-26a-5p and IL-6 and ICAM-1. These data implicate miR-29b-3p and miR-26a-5p loss as the cause or consequence of inflammation in COVID-19, whilst the relevance of miR-34a-5p downregulation is unclear.

The study does not explore the contextual mechanism of these miRNAs, but they may be further evidence of the unique infection and CSS-related inflammation that is severe COVID-19, given that existing literature implicates these two miRNAs in both pro-inflammatory and anti-inflammatory function, depending on disease. For instance, in a model of particulate matter-induced respiratory disease (therefore not infection-related), miR-29-3p was in fact found to promote IL-8 and other pro-inflammatory cytokine expression [41], despite being inversely correlated with IL-8 in COVID-19 [37]. Conversely, in an infection event, another study linked miR-29b-3p upregulation with reduced MAPK activation and NF-κB signalling after LPS stimulation in neonatal rat cardiomyocytes, reducing inflammatory damage in this model of sepsis-induced cardiac arrest [40]. As such, miR-29-3p loss may be contributory to hyperinflammation solely in states of infection-related immune dysregulation. Further, miR-26a-5p increases correlated with increased expression of IL-1β, IL-6, and TNF-α in macrophages from diabetic mice [39], conflicting with the observation in the COVID-19-related study that miR-26a-5p is inversely correlated with IL-6 expression [37]. However, in a lung infection and injury model using LPS, miR-26a-5p overexpression improved immunopathological disease markers, likely via the observed reduction of TNF-α and IL-6. These comparisons therefore highlight COVID-19 as a novel but infection-related disease in its ncRNA signature. The results of study [37] further suggest that ncRNA regulation should be placed within the context of specific tissues and disease contexts, therefore justifying further pursuit of their specific mechanistic involvement in COVID-19 pathogenesis.

## 5. ncRNAs in COVID-19-Associated Haemostatic Dysregulation

### 5.1. Venous Thromboembolic Events in COVID-19

The clinical term venous thromboembolic event (VTE) groups incidences of deep vein thrombosis (DVT), the formation of a vascular clot usually in the major calf muscles, and the possible associated severe complication of pulmonary embolism (PE), the displacement of a clot and its subsequent obstruction of a lung artery, together [85]. Thromboembolic events and haemostatic dysregulation are increasingly regarded as core pathogenic outcomes of COVID-19 and a substantial cause of death. A combinatorial analysis of 102 studies on a collective 64,503 patients ascertained that between 14.7% and 17.6% of those hospitalised with COVID-19 experienced a VTE [86]. Whilst studies continue to extract the mechanisms surrounding this surprisingly prevalent side-effect from a viral infection, ncRNAs are continually implicated.

### 5.2. COVID-19 Patients with High Coagulability Have Deregulated Inflammatory miRNAs

The differential expression of miRNAs has also been connected to COVID-19 vascular pathology by stratifying patients with COVID-19 based on D-dimer levels [42]. D-dimer is a serum-measurable product of fibrin degradation and is thus indicative of coagulation and fibrinolytic activity [87]. In the circulating exosomes of high D-dimer COVID-19 patients, there was significant downregulation of miR-103a, miR-145, and miR-885, whilst miR-424 was significantly upregulated [42]. The first-mentioned miRNA, miR-103a, was also downregulated in another study of non-COVID-19 DVT patients compared to healthy controls, and intriguingly, this relationship was also noted in a subsequent mouse model [43]. This suggests that, via this specific miRNA, COVID-19 patients exhibit common VTE-related ncRNA dysregulation despite the difference in aetiologies.

In the same COVID-19 study, miR-145 was downregulated [42]. One of the predicted targets of miR-145 is the tissue factor (TF) [45]. TF, once bound to factor VII/VIIa, initiates coagulation and is thus a key component in clot formation [46], and consequently, its regulation is of high relevance in clot-related disease. The restoration of miR-145 expression in a thrombotic animal model decreased TF activity and serum levels, which was subsequently associated with reduced thrombogenesis [46]. Levels of miR-145 were also reduced in non-COVID-19 VTE patients and were negatively correlated with TF levels [46].

The third miRNA, miR-885, is thought to target the von Willebrand factor (vWF), another key component of the clotting cascade [47]. Although predominantly characterised for its role in arterial thrombosis, together with a-disintegrin-like-metalloprotease-with-a-thrombospondin-type-1-motif-member 13 (ADAMTS13), vWF is also directly implicated in VTE and is discussed accordingly as a therapeutic target [88]. Interestingly, the vWF/ADAMTS13 axis is directly investigated in the context of COVID-19-related VTE [48]. ADAMTS13 activity and vWF:Ag levels are correlated with VTE incidence in that markedly high vWF concentrations are associated with lower ADAMTS13 activity, and interestingly, also with high D-dimer levels. Given these results presented in the miRNA COVID-19 study [42], miR-885 levels would expectedly be low in the high-D-dimer COVID-19 patients of the vWF/ADAMTS13 study [48], placing miR-885 fascinatingly within the context of VTE incidence due to vWF/ADAMTS13 imbalance, and thus higher mortality in these patients.

### 5.3. A High D-Dimer Specific miRNA Predicts Thromboembolic Events

Finally, the study uncovered the COVID-19 high D-dimer-specific upregulation of miR-424 [42]. In terms of VTEs, this is a comparatively more researched miRNA, as its upregulation has been noted in high-D-dimer patients in two previous studies [49,50]. It may also promote monocyte differentiation [51] that is suggestive of inflammatory innate immune cell proliferation. In the study, the expression of miR-424 directly predicted thromboembolic events in COVID-19 patients, suggesting its use as a biomarker to stratify severe COVID-19 patients into those who may be prone to a VTE [42]. In particular, this may be the case in a context where VTE is a common and potentially lethal disease element of COVID-19, but its occurrence is not always noted.

## 6. ncRNAs in COVID-19-Associated Vascular Dysregulation

### 6.1. miRNAs and Cardiovascular Disease in COVID-19

SARS-CoV-2 infection is instigated by the inhalation of virions into the lung and their subsequent attachment and entry via the ACE2 receptor, which is highly expressed by respiratory tract cells [3]. However, other human cell types similarly express this receptor, and thus are potential hosts for SARS-CoV-2, such as arterial and venous endothelial cells, epithelial cells of the small intestine [4], and prominently, cardiac cells [5]. Cardiac alterations are a complication of COVID-19 and commonly present in severe patients [1,89]. As such, research has been directed towards cardiovascular regulation-related miRNAs and their potential participation in this pathology. Four miRNAs; miR-21, miR-155, miR-208a, and miR-499 were found to be upregulated in the sera of patients with severe COVID-19 [52]. The inflammatory relevance of miR-155 was previously discussed in Section 3.4. Otherwise, of particular interest was the COVID-19-specific upregulation of miR-499, as the study looked at expression of their miRNAs in a separate cohort COVID-19 patients compared to patients with influenza-induced ARDS. The relevance of COVID-19-specific miR-499 upregulation is unclear; however, it is an established regulator of cardiac muscle fibre identity and thus contractility, by promoting the development of fast or slow myosins [57].

Targeted assessment of miR-21-5p expression in COVID-19 patients was justified in the study [52] by its previously established relationship with cardiovascular regulation. It is widely expressed across cardiac tissues and is notably aberrantly expressed in cases of heart-related disease [53,90]. For instance, a mouse model of cardiac hypertrophy exhibited a four-fold increase in expression of miR-21 compared to sham surgery mice [53]. Further, antisense-mediated depletion of miR-21 in cultured neonatal cardiomyocytes showed improvement of an induced hypertrophic state [53]. Similarly, miR-21 expression was shown to potentiate ERK-MAPK activity by inhibiting sprouty homologue 1, causing cardiac fibrosis and dysfunction [54]. These effects were reversed when miR-21 was silenced. However, conflicting results have again been published that emphasise the need to place specific miRNAs within cell, tissue, and disease-specific contexts to understand their holistic effect. For example, the overexpression of miR-21 in mice with cardiac infarctions improved symptoms and reduced fibrosis [55].

In a similar narrative, miR-208a is encoded from an intron in the α-myosin heavy chain gene, the product of which is a cardiac contractility protein, meaning that the expression of this miRNA is cardiac-specific [58]. Interestingly, miR-208 knockout mice did not develop hypertrophy or fibrosis in response to thoracic aortic banding, a procedure which leads to a hypertrophic model of heart disease in wild-type mice [57]. The study uncovered that miR-208 expression upregulates the β-myosin heavy chain gene in stress-specific contexts [57], an effect which has been shown to promote cardiac remodelling and fibrosis [91]. Altogether, the data suggest that a COVID-19-induced, but not COVID-19-specific, state of miRNA-related cardiac dysregulation may exist in severely affected patients. This state may be readable through miR-21 and miR-208a expression levels in sera, and these miRNAs may also play a direct mechanistic role in COVID-19-related cardiac pathogenesis.

### 6.2. miRNAs and Cerebrovascular Disease in COVID-19

Similarly, focus has been directed towards the cerebrovascular (CBV) complications of COVID-19, for instance, stroke and ischaemic attack, and once again, ncRNAs are implicated. In one of the largest patient profiling studies published to date, the authors investigated the miRNA content of extracellular vesicles in the plasma of 321 COVID-19 patients and attempted the identification of correlates with COVID-19-specific CBV events [92]. They accordingly uncovered the significantly associated downregulation of miR-24, which, interestingly, did not occur in uninfected patients suffering from a CBV event. Prior studies had identified neuropilin-1 as a target of miR-24, and indeed, that neuropilin-1 is a necessary co-factor of SARS-CoV-2 infection and is associated with COVID-19-associated CBV events. To elegantly complete the narrative to date, miR-24 is specifically suggestive of increased neuropilin-1 expression in human brain endothelial cells [93]. Therefore, miR-24 may not be only highly useful as a predictor of CBV events, it may also be understood as a participant in the COVID-19-specific pathogenesis of brain-related vascular dysregulation.

## 7. Interactions between Host and SARS-CoV-2-Derived ncRNAs

### 7.1. The SARS-CoV-2 RNA Genome as a Non-Coding Interactor with Host RNA

The genome of SARS-CoV-2 is itself single-stranded RNA and therefore can putatively interact with host miRNAs upon infection and uncoating whilst also likely producing ncRNAs itself. Interactions could be beneficial for the host due to the inhibition of viral protein translation or replication, but this may only take place in undifferentiated stem cells [94]. Equally, viral RNA may negatively alter the regulatory activity of host miRNAs via direct binding, changing behaviour and signalling. SARS-CoV-2 may itself produce miRNAs with host-altering function, as it has been demonstrated for its sister virus, SARS-CoV [95]. One study utilized a computational approach to predict possible miRNAs within the SARS-CoV-2 genome and identified viral-derived miRNAs that could alter interferon, WNT, and mTOR signalling as well as autophagy [96]. Another study, currently in preprint, again used computational prediction to identify a SARS-CoV-2-derived miRNA dubbed MD147-3p with the sequence CCCUGAUGAGGGUGGGUUC [97]. Intriguingly, the authors predict that this miRNA targets the enhancer of TMPRSS2, the serine protease responsible for spike protein cleavage, a precursor to SARS-CoV-2 cell entry. Further, an integrated computational analysis of the interactions predicted between host and SARS-CoV-2 ncRNA and protein suggest that crosstalk between these components may alter a key immunological pathway, TGF-β signalling [98].

These concepts were explored in a study in which the authors started by infecting cultured cell lines with SARS-CoV-2 [99]. RNA sequencing was then used to identify a swathe of virus-derived small RNAs. Amongst the top ten most expressed small RNAs was one dubbed v-miRNA-N-28612, which mapped to the nucleocapsid (*N*) gene. Interestingly, a translational study in COVID-19 patients uncovered that the relative expression levels of v-miR-N-28612 correlated with the viral load in nasopharyngeal samples [99]. Additional bioinformatic work then determined that this miRNA, amongst the others identified, is likely to target transcripts associated with cell metabolism and biosynthesis, suggesting that SARS-CoV-2 infection-induced changes to cell behaviour may be specifically mediated by virus-derived ncRNAs. Finally, the overexpression of v-miRNA-N-28612 in PBMCs in cell culture upregulated IL-1β, caspase 1, and NLRP3, which are markers of the inflammasome, suggesting that SARS-CoV-2 ncRNAs can directly mediate hyperinflammation, a hallmark of COVID-19.

### 7.2. Differences in miRNA Targeting May Explain Divergent SARS-CoV-2 Virology

In additional computational work, one study explored the putative host miRNA targeted sites in the RNA genomes of SARS-CoV-2, SARS-CoV, and Middle East respiratory syndrome CoV (MERS-CoV) as well as non-pathogenic CoVs [100]. Generally, the data showed that pathogenic strains had relatively more putative miRNA binding sites compared to those that were non-pathogenic. Whilst the specific outcomes of this characteristic are unclear, one suggestion is that this allows the viral genome to deplete host miRNAs to favour viral propagation. For instance, the miRNAs predicted to selectively target pathogenic strains tended to be associated with cellular processes such as apoptosis, cytoskeletal remodelling, and antigen presentation. Specifically, 28 miRNAs were predicted to uniquely target SARS-CoV-2, which were found to be well expressed in bronchial epithelial cells and that were also found to be dysregulated in instances of disease, such as lung cancer, chronic obstructive pulmonary disorder, and cystic fibrosis.

### 7.3. Host and Viral RNA-Binding Proteins and ncRNAs as Regulators of Infection

A class of proteins whose function and regulation are intrinsically linked with the behaviour and characteristics of RNA are RNA-binding proteins (RBPs). Indeed, the interconnected crosstalk between ncRNAs and RBPs can have drastic impacts on cell behaviour. For instance, the cognate antisense lncRNA from the sixth intron of the metastasis-associated in colon cancer-1 gene (MACC1-AS1) can mediate the characteristics of breast tumour progression [101]. MACC1-AS1 acts as an endogenous competitor for the miRNAs miR-384 and miR-145-3p, abrogating their capacity to bind to their respective target mRNAs that encode c-Myc and pleiotrophin. Interestingly, MACC1-AS1 can bind the RBP polypyrimidine-tract-binding protein 1 (PTBP1), enhancing its ability to act as a competing endogenous RNA whilst pulling PTBP1 from its target mRNAs in parallel, changing alternative splicing patterns with a ‘sponging’ effect.

This conceptual understanding representing the crossover between RBPs and ncRNA is also represented in COVID-19 research, where researchers have investigated the effects on the host cell RNA-bound proteome after infection. A recent study reported that not only does a third of the entire host RBPome remodel during infection, but that the pharmacological inhibition of specific cellular RBPs such as HSP90 and IGF2BP1 can stall the production of SARS-CoV-2 infection-derived viral proteins, therefore hampering replication [102]. It was also found that RBPs that constitute the tRNA ligase complex are also components of the SARS-CoV-2 viral ribonucleoprotein—their knockdown in cell experiments reduced the burden of intracellular viral RNA [102]. Similarly, one study identified that 13 of the 29 proteins encoded by the SARS-CoV-2 genome likely interact with a corresponding 51 human-encoded RBPs and that an according shift in alternative splicing patterns occurs in infected lung cells [103]. In addition, bioinformatic analysis identified an enrichment for potential binding sites for host RBPs in the SARS-CoV-2 genome, suggesting that a sponging effect may occur that is similar in mechanism to MACC1-AS1 and PTPB1 [103].

## 8. Concluding Remarks

Though a majority of cases present as a mild viral infection of the lung, COVID-19 can manifest in interlinked multi-system dysregulation that underlies a relatively high mortality rate in specific patient populations, initiating a cytokine storm condition of hyperinflammation, inflicting organ damage through excessive immune cell recruitment, and dysregulating the haemostatic and cardiovascular systems [3,6,7,8,9]. Despite the infancy of the field, the unique pressure of a global pandemic has pushed research forward rapidly, motivating the extensive pursuit of in-depth mechanistic understanding of immunity and disease development following infection. A multitude of studies have now implicated ncRNAs in most aspects of COVID-19 disease. Collectively, these studies converge on the mechanistic and clinical significance of ncRNAs in COVID-19.

A hugely useful outcome of this research would be the establishment of biomarkers to provide informative and easily accessible readouts for patient stratification. This is not only in the context of predicting outcomes in hospitalised individuals, but also responses to vaccination and risk factors associated with long-term COVID-19. Many of the already-known disease hallmarks of COVID-19 have been shown to be associated with circulating serum concentrations of specific miRNAs, for instance miR-424, in the likelihood of a thromboembolic event [42], and miR-766-3p in its association with IL-6-related hyperinflammation [19]. Further, the expression of multiple ncRNAs could be integrated to provide a detailed picture of disease. For instance, the exploration of three miRNAs distinguished SARS-CoV-2 infection in humans and in an animal model with over 99.7% accuracy [19]. Moreover, the utility of serum miRNA measurement may extend to monitoring and predicting the likelihood of therapeutic success [25]. The primary change necessary for clinical implementation would be consistent results from larger prospective studies, as studies up until now have focused on retrospective analyses of relatively small patient cohorts.

The study of ncRNAs is also key to gaining insight into the mechanisms that drive severe COVID-19. For instance, the intriguing crosstalk between the lincRNA PVT1, the miRNAs miR-146a and miR-155 as well as miR-766-3p in NF-κB signalling and their substantial relevance in the lethal cytokine storm syndrome. Further, the fascinating notable downregulation of NEAT1 across immune cells in mild patients serves as early evidence of host protection, specifically in the context of immune cell recruitment and damaging effector function. In addition, understanding and appreciating the COVID-19-related miRNA expression changes affecting haemostasis and cardio/cerebrovascular regulation could be crucial in combatting these commonly serious peripheral disorders. The use of appropriate pre-clinical models of COVID-19 and experimental medicine studies in humans will be critical for dissecting the role of ncRNAs in immunity to SARS-CoV-2 and COVID-19 pathology.

Given SARS-CoV-2 is itself composed of RNA with interacting potential, work has also been directed towards predicting or investigating the possibility of host or viral-derived ncRNA interactions, to the detriment of the virus or, indeed, the host. Further work in this area may cultivate an area of developing drug discovery, in disrupting RNA-RNA or RNA-protein interactions, to mitigate viral propagating mechanisms such as miRNA and RBP sponging, or to limit host hyperinflammation from viral ncRNA expression. COVID-19 may become endemic in years to come [104]; therefore, discovery of novel drug targets will be vital in improving disease outcomes. In this respect, unlocking the untapped potential of the non-coding transcriptome as a source of potentially druggable targets can be transformative in our search for COVID-19 therapeutics.

Altogether, the data presented herein and throughout the literature highlight a crucial role for ncRNAs in COVID-19. Further research is necessary to establish clinically usable miRNA signatures, to investigate ncRNA-based immunological and peripheral system regulation, to explore the importance of SARS-CoV-2 as a source of interacting RNA itself, and to uncover novel drug targets in this unprecedented disease.

## Figures and Tables

**Figure 1 ncrna-07-00054-f001:**
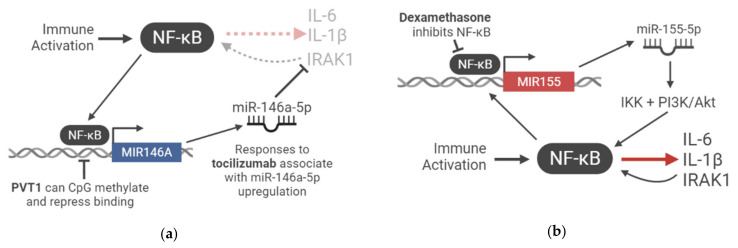
NF-κB amplification of the immune-activating signal is an integration of ncRNA regulation. (**a**) A negative feedback loop of immunosuppressive ncRNA regulatory balance: NF-κB signaling induces the transcription of miR-146a-5p, which inhibits IRAK1, a positive regulator of NF-κB-promoted genes. The result is the downregulation of inflammatory gene expression, such as IL-6, IL-1β, and IRAK1. The lincRNA PVT1 is associated with promoter CpG methylation and thus the downregulation of miR-146a-5p expression, blocking its immunosuppressive expression; (**b**) a positive feedback loop of inflammatory ncRNA regulatory balance: NF-κB signaling induces the transcription of miR-155-5p, which further amplifies NF-κB signaling via IKK and PI3K/Akt, leading to the production of the inflammatory molecules IL-6, IL-1β, and IRAK1. Dexamethasone treatment can block miR-155-5p transcription by inhibiting NF-κB signalling.

**Table 1 ncrna-07-00054-t001:** A summary of the published COVID-19 ncRNA profiling studies with relevant literature highlighted.

Study	ncRNA of Interest	Effect on ncRNA from Infection	Context Investigated	Additional Findings
[19]	miR-766-3p	Downregulated	COVID-19 patients	▪ miR-766-3p was also downregulated in arthritis patients where patients upregulated IL-6 [20]. ▪ miR-766-3p overexpression reduced IL-6 expression in a cell line-immune stimulation model [21].
	miR-1275	Downregulated	COVID-19 patients	▪ TNF-α and IL-6 treatment of adipocytes reduced miR-1275 expression. ▪ miR-1275 has NF-κB binding sites.
	miR-31-5p	Upregulated	COVID-19 patients	▪ miR-31 was overexpressed in keratinocytes from patients with psoriasis (inflammatory condition) [22]. ▪ miR-31 silencing also suppressed the ability of keratinocytes to recruit immune cells. ▪ miR-31 was found to inhibit STK40, an immunosuppressive protein. ▪ However, miR-31 is also associated with TNS1 downregulation, an enhancer of immune infiltration [23]. ▪ Additionally, miR-31 knockdown worsened inflammation in a mouse inflammatory bowel disease model [24].
[25]	miR-146a-5p	Downregulated	COVID-19 patients	▪ miR-146a-5p was upregulated in patients that responded to tocilizumab treatment. Those who were unresponsive downregulated it [25]. ▪ miR-146a-5p was found to downregulate IL-6 expression [26]. ▪ miR-146a-5p was also found to downregulate IL-1β and IRAK1 expression [27].
[28]	PVT1	Upregulated	COVID-19 patients	▪ PVT1 promotes the CpG methylation of the miR-146a promoter, suppressing expression [29]. ▪ PVT1 was also upregulated in the synovial tissue of arthritis patients [30].
[31]	MALAT1	Upregulated	SARS-CoV-2 infected NHBE cells (bronchial)	▪ MALAT1 can enhance immune cell chemotaxis by recruiting p300, reducing IL-8 expression [32].
[33]	MALAT1	Downregulated	Mild COVID-19 patients—monocytes and macrophages	▪ MALAT1 expression is associated with macrophage differentiation into the inflammatory M1 subtype [34]. These mildly affected patients may be exhibiting protection [33].
	MALAT1	Upregulated	Mild COVID-19 patients–CD4+ T cells	▪ MALAT1 loss activates CD4+ T cells, pushing the balance away from regulatory T differentiation, instead towards the Th1 and Th17 effector type [35]. ▪ MALAT1 mouse knockouts have more immune activation in infection [17].
	MALAT1	Downregulated	Severe COVID-19 patients—CD4+ T cells	
	NEAT1	Downregulated	Mild COVID-19 patients—BAL cells	▪ NEAT1 enhances the assembly and processing of inflammasomes in macrophages; thus, its expression is likely pro-inflammatory [36].
	NEAT1	Upregulated	Severe COVID-19 patients—BAL cells	
[37]	miR-26a-5p	Downregulated	COVID-19 patients	▪ miR-26a-5p downregulation correlated with IL-6 and ICAM-1 upregulation in the study patients [37]. ▪ miR-26a-5p overexpression improved lung disease in an LPS-induced infection mouse model, likely by reducing inflammatory cytokine expression [38]. ▪ However, miR-26a-5p is correlated with increased IL-1β, IL-6, and TNF-α expression in macrophages from diabetic mice [39].
	miR-29-3p	Downregulated	COVID-19 patients	▪ miR-29-3p downregulation is correlated with IL-4 and IL-8 upregulation in the study patients [37]. ▪ miR-29-3p acted as anti-inflammatory by reducing MAPK activation and NF-κB signalling after LPS stimulation in a rat model of sepsis [40]. ▪ However, miR-29-3p was thought to promote IL-8 and other cytokine expression in a mouse respiratory disease model [41].
[42]	miR-103a	Downregulated	Higher D-dimer COVID-19 patients	▪ miR-103a was downregulated in another study of patients with thromboembolic events [43]. ▪ miR-103a promotes M2 polarization, an immunosuppressive macrophage subtype [44].
	miR-145	Downregulated	Higher D-dimer COVID-19 patients	▪ miR-145′s predicted target is the tissue factor (TF) [45]. ▪ Restoring miR-145 in a thrombotic animal model decreased TF and reduced thrombogenesis [46]. ▪ miR-145 was also reduced in patients with thromboembolic events and was negatively correlated with TF levels [46].
	miR-885	Downregulated	Higher D-dimer COVID-19 patients	▪ miR-885′s predicted target is the von Willebrand Factor (vWF) [47]. ▪ ADAMTS13/vWF is correlated with thromboembolic incidence in COVID-19, and higher vWF is correlated with high D-dimer levels [48]
	miR-424	Upregulated	Higher D-dimer COVID-19 patients	▪ miR-424 was also upregulated in two other studies of thromboembolic patients [49,50]. ▪ miR-424 may promote monocyte differentiation [51].
[52]	miR-21	Upregulated	COVID-19 patients	▪ A mouse model of cardiac hypertrophy exhibited a four-fold increase in miR-21 [53]. ▪ Antisense miR-21 depletion in cultured heart cells improved their hypertrophic state [53]. ▪ miR-21 potentiated ERK–MAPK activity by inhibiting sprout homologue 1, inducing cardiac fibrosis and dysfunction [54]. ▪ However, miR-21 overexpression improved fibrosis and symptoms in mice with cardiac infarctions [55].
	miR-155	Upregulated	COVID-19 patients	▪ miR-155 activates NF-κB signaling by activating IKK and PI3K/Akt [56]
	miR-208a	Upregulated	COVID-19 patients	▪ miR-208a is cardiac-specific [57]. ▪ miR-208 knockout mice did not develop fibrosis or hypertrophy in a heart disease model [58].

## Data Availability

Not applicable.

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
