# Peer review of "Non-Coding RNAs in COVID-19: Emerging Insights and Current Questions"

_ncrna, 2021, doi:10.3390/ncrna7030054_

Round 1

Reviewer 1 Report

Authors have scripted a timely review on “Non-coding RNAs in COVID-19: Emerging insights and current questions” that provide a broad perspective on the involvement of short and long ncRNAs in COVID-19 with recent research evidence. While the article in its current state does cover the required evidence and discussion on short and lncRNAs in context to SARS-CoV-2, it will be imperative to discuss more on their cross talk and role in the host’s post-transcriptional regulation and alteration during COVID-19 illness. I personally like the section, “Interactions between host and SARS-CoV-2-derived ncRNAs” and follow up section, which I believe should strengthen the review further if the authors put their rationale (based on literature in general, not necessarily related to COVID-19) on how the host short and long ncRNAs mutually interact/ tolerate the virus (SARS-CoV-2). I think this will help to comprehend the importance of ncRNAs and their cross-talk/ post-transcriptional regulation in COVID-19 patients, explaining the titration of the virus by miRs/ lncRNAs alongside being sponged by RBPs (PMIDs: 31822653, 32993015, etc can be useful).

Authors can optionally refer to the below articles, which I believe can help to elaborate the requested section.

https://www.nature.com/articles/s41598-021-86134-0
https://www.meta.org/papers/the-crosstalk-between-bone-metabolism-lncrnas/32971215

https://www.ncbi.nlm.nih.gov/pmc/articles/PMC7386606/

https://pubmed.ncbi.nlm.nih.gov/33670580/

Author Response

We thank the Reviewer for the helpful suggestions. We have expanded section 7 as suggested and addressed further cross-talk between ncRNAs.

Reviewer 2 Report

This manuscript highlights the involvement of multiple ncRNAs (both short and long ncRNAs) in the immunopathology of COVID-19. Overall, the manuscript is well written and gathers a large breadth of information on the involvement of ncRNAs in COVID-19. Table 1 is quite informative and clear. However, contrasting findings between studies for multiple ncRNAs and their often unclear mechanism of action make it easy to disregard the importance of ncRNAs in disease pathogenesis and their clinical utility. I would have liked to see a stronger conclusion section that tied everything together and highlighted the potential for ncRNAs in clinical case management as well as the type of studies needed to support this.

Additionally, some specific concerns:

  • While well written, at times the manuscript seems to lose focus at bit. For example, lines 155-167 are clearly very relevant and the data presented here is interesting, but these paragraphs feel awkward and leave the reader a bit unclear as the focus of the subsection is stated to be miR-766-3p.  
  • Line 503-507 – citation?
  • Line 536 – while many scientists expect COVID-19 to become endemic, I didn't think we are there yet. Citation?

Author Response

We thank the Reviewer for the helpful suggestions.

We have amended the text to discuss some potentially contradicting results. We have also amended our concluding paragraph as suggested.

We have amended the text on miR-766-3p. 

We have now added the requested references and used more careful language when stating that COVID-19 may become endemic.